# Effect of Vitamin-D-Enriched Edible Mushrooms on Vitamin D Status, Bone Health and Expression of CYP2R1, CYP27B1 and VDR Gene in Wistar Rats

**DOI:** 10.3390/jof8080864

**Published:** 2022-08-17

**Authors:** Muneeb Ahmad Malik, Yasmeena Jan, Lamya Ahmed Al-Keridis, Afrozul Haq, Javed Ahmad, Mohd Adnan, Nawaf Alshammari, Syed Amir Ashraf, Bibhu Prasad Panda

**Affiliations:** 1Department of Food Technology, School of Interdisciplinary Science and Technology, Jamia Hamdard, New Delhi 110062, India; 2Department of Biology, College of Science, Princess Nourah bint Abdulrahman University, P.O. Box 84428, Riyadh 11671, Saudi Arabia; 3Department of Biotechnology, School of Chemical and Life Sciences, Jamia Hamdard, New Delhi 110062, India; 4Department of Biology, College of Science, University of Hail, P.O. Box 2440, Hail 55476, Saudi Arabia; 5Department of Clinical Nutrition, College of Applied Medical Science, University of Hail, P.O. Box 2440, Hail 55476, Saudi Arabia; 6Microbial and Pharmaceutical Biotechnology Laboratory, Department of Pharmacognosy and Phytochemistry, School of Pharmaceutical Education and Research, Jamia Hamdard, New Delhi 110062, India

**Keywords:** serum 25-OHD, vitamin D deficiency, ultraviolet irradiation, bioavailability, gene expression

## Abstract

Vitamin D deficiency is highly prevalent in India and worldwide. Mushrooms are important nutritional foods, and in this context shiitake (*Lentinula edodes*), button (*Agaricus bisporus*) and oyster (*Pleurotus ostreatus*) mushrooms are known for their bioactive properties. The application of ultraviolet (UV) irradiation for the production of substantial amounts of vitamin D_2_ is well established. Levels of serum 25-hydroxy vitamin D (25-OHD), parathyroid hormone (PTH), calcium, phosphorus and alkaline phosphatase (ALP) were significantly (*p* < 0.05) improved in vitamin-D-deficient rats after feeding with UVB irradiated mushrooms for 4 weeks. Further, microscopic observations indicate an improvement in the osteoid area and the reduction in trabecular separation of the femur bone. In addition, the level of expression of the vitamin D receptor (VDR) gene and genes metabolizing vitamin D were explored. It was observed that in mushroom-fed and vitamin-D-supplemented groups, there was upregulation of CYP2R1 and VDR, while there was downregulation of CYP27B1 in the liver. Further, CYP2R1 was downregulated, while CYP27B1 and VDR were upregulated in kidney tissue.

## 1. Introduction

Vitamin D deficiency has emerged as a common metabolic disorder globally [1]. Generally, vitamin D deficiency is described as a serum 25-OHD concentration of <20 ng/mL (50 nmol/L), which is a cut-off point from where parathyroid hormone starts to rise, which is the physiological definition of vitamin D deficiency [2]. In India, vitamin D deficiency is estimated in about 50–90% [3]. In the United States, 30% of the population and in Europe 40% of the population are reported to have vitamin D deficiency [2,4]. In Africa, more than 50% are vitamin D deficient [5]. Some Middle East countries such as the United Arab Emirates and Saudi Arabia also report high levels of vitamin D deficiency (50% and 59% respectively) despite having abundant sunshine [6].

Vitamin D is found in two main forms, “cholecalciferol” (vitamin D_3_) and “ergocalciferol” (vitamin D_2_), which are structurally similar with the exception of a methyl group on carbon 24 and a double bond between carbons 22 and 23 [7]. Dietary sources of vitamin D_3_ include animal sources, while dietary sources of vitamin D_2_ are phytoplankton, invertebrates and fungi [8]. The principal source of vitamin D_3_ for nearly all people is UV-B (290–315 nm) irradiation from direct sunlight, which triggers vitamin D formation in the skin [9]. In modern times humans tend to stay indoors, and while moving outdoors they preferentially cover their large skin surface with clothing, resulting in insufficient exposure to the sun’s UV-B and consequently low production of endogenous vitamin D [10]. Dietary sources of vitamin D are limited only to some fishes or cod liver oil. Some countries carry out vitamin D fortification of certain limited food products such as milk, orange juice and some cereals and yogurts [11]. However, these efforts are not enough to tackle vitamin D deficiency; therefore, alternate ways are sought to meet the needs of vitamin D [12]. Among the well-known sources of vitamin D, edible mushrooms are the only vegetarian source, popular for their delicacy, nutritional and therapeutic values among both vegetarians and non-vegetarian population groups. The fact is that cultivated mushrooms possess little or no vitamin D_2_, but they are rich in the vitamin D_2_ precursor “ergosterol” [13]. Enrichment of vitamin D in mushrooms can be achieved by exposing them to ultraviolet light energy (UV-A, UV-B and UV-C), which converts ergosterol to ergocalciferol/vitamin D_2_ [7].

Several studies have been carried out in past years to measure the effects of ultraviolet irradiation on vitamin D_2_ enrichment in several mushroom species and have reported the vast potential of UV-exposed mushrooms as a source of vitamin D_2_. Most of the previous research papers have been mainly centered on commercially grown species such as *Lentinula*, *Agaricus* and *Pleurotus* sp. These mushroom species have high contents of nutritional and bioactive components that can be employed as a bio-based multifunctional ingredient to lower vitamin D deficiency and promote other health benefits. Mushrooms are reported to provide a comprehensive healthy diet, possessing all the essential nutrients and several nutraceutical compounds such as polysaccharides, terpenes and biologically active proteins. In addition to this, they are also rich in various beneficial compounds such as β-glucans, tocopherols, terpenoids, lectins, etc., imparting them with strong therapeutic effects for preventing and curing many degenerative diseases [14,15]. Vitamin-D-rich mushrooms are deemed to be the next-generation functional food because of their potential as a remedy for many degenerative diseases such as osteoporosis, diabetes, cancers, cardiovascular diseases, etc. [16]. To the best of our knowledge, very few attempts have been made in the past to study the biological effects of vitamin D_2_ from UV-irradiated mushrooms. Two previous in vivo studies confirmed the bioavailability vitamin D_2_ from UV-B irradiated shiitake [17] and white button mushrooms [18]. However, little is known about additional health-related benefits and the efficacy of UV-irradiated mushrooms to increase serum 25-OHD levels.

Vitamin D obtained from sunlight or diet is converted in the liver to 25-hydroxyvitamin D by 25-hydroxylase (CYP2R1), and this form of vitamin D is the major circulating form and marker of serum vitamin D levels in the body. However, it is not biologically active and is converted by 1α-hydroxylase (CYP27B1) to its active form 1,25-dihydroxy vitamin D in the kidney (Figure 1). The vitamin D receptor protein, encoded by the VDR gene, mediates the biological functions of vitamin D. The VDR protein binds to the active form of vitamin D, which allows it to interact with the retinoid X receptor protein. The resulting heterodimer then binds with vitamin D response elements in target genes and regulates their expression. It has been predicted that more than 200 genes are regulated by the active form of vitamin D, influencing a variety of functions including calcium levels, phosphate absorption and other cellular processes [19].

Thus, the present study was aimed at evaluating the efficacy of vitamin-D-enriched mushrooms in improving blood biochemical parameters and bone homeostasis in vitamin-D-deficient rats and equating it with marketed supplements of vitamin D_2_ and D_3_. The effect on the level of expression of vitamin-D-related genes (CYP2R1, CYP27B1 and VDR) in the liver and kidney of rats was also evaluated.

## 2. Materials and Methods

### 2.1. Mushrooms and Vitamin D Supplements

Fresh shiitake, button and oyster mushrooms were obtained from a local mushroom farm (Indian Organic Farming) located in Haryana, India. Supplements of vitamin D_2_ (Lalmin, 8000 IU/g or 200 µg/g, batch no: 804355E) were gifted from Lallemand, USA, and vitamin D_3_ (Calcirol, 60,000 IU/g or 1500 µg/g, batch no: JK142014) manufactured by Cadila Pharmaceuticals Ltd., India, was purchased from a local medical store in New Delhi, India.

### 2.2. Irradiation of Mushrooms and Vitamin D_2_ Analysis

The optimum conditions for vitamin D_2_ enrichment of edible mushrooms derived from our previous study were applied [20]. Four main factors (moisture content, type of wavelength, distance between UV source and mushroom trays and duration of UV exposure) that can affect vitamin D_2_ synthesis were selected. The moisture content of mushrooms was reduced to 76–78% using a tray drier (60 °C), and the mushrooms were subjected to UVB (290–315 nm) irradiation at a distance 20–30 cm from the UV light source for 120 min. After irradiation, mushrooms were lyophilized and ground to powder.

The content of the vitamin D_2_ in the irradiated mushrooms was determined as described by Malik et al. [20]. Briefly, 2 g lyophilized powder as weighed and transferred into 50 mL centrifuge tubes and suspended in 50% potassium hydroxide (5 mL) and DMSO (5 mL). After vortex mixing, the mixture was ultrasonicated (PCI Analytics Pvt Ltd., Mumbai, India) for 30 min. Afterwards, 20 mL of n-hexane was poured, mixed well and again sonicated for 30 min. The mixture was then centrifuged at 5000× *g* for 8 min at 25 °C (Remi centrifuge, India), the supernatant (n-hexane layer) was collected and the extraction process was repeated two times. The organic layer was then transferred into a 100 mL round bottom flask for complete drying by rotary evaporator. The sample was then immediately redissolved in 1 mL of HPLC grade methanol and filtered using a 0.45 µm membrane filter. HPLC was performed using an isocratic HPLC system (Shimadzu corporation) fitted with a SIL-20AC-HT auto sampler, DGU-20A 5R degassing unit and LiChrospher 100 RP-C18 column (250 × 4.6 mm × 5 mm). Composition of the mobile phase was methanol and acetonitrile (80:20), and the flow rate was 1 mL/min with a total run time of 20 min. Detection was carried out using a PDA detector at 264 nm. The vitamin D_2_ content in the mushrooms was calculated based on the calibration curve. The vitamin D_2_ content in UV-irradiated shiitake, button and oyster mushrooms was 25.6 ± 0.42, 32.4 ± 1.45 and 28.7 ± 1.14 μg/g, respectively.

### 2.3. Animals

For the in vivo assay, 36 Wistar albino rats (both genders), 4–6 weeks old, with mean body weight (BW) of 74 ± 13 g, were obtained from the central animal house facility, Jamia Hamdard University, New Delhi, India. The animals were housed in polyethylene boxes and kept at room temperature (25 ± 2 °C) and received water and commercial feed ad libitum. The animals were first acclimatized for a week before the experiment. After approval from the Institutional Animal Ethics Committee, Jamia Hamdard (registration no. 173/GO/Re/S/2000/CPCSEA, proposal no 1557/2019), the experiments were conducted by following the guidelines of the Committee for the Purpose of Control and Supervision of Experiments on Animals (CPCSEA), Govt of India.

The rats were shifted to covered cages to limit direct exposure to fluorescent light and were fed on a customized, vitamin-D-deficient diet (VDDD) (Custom AIN-93G, Bio-Serve, NJ, USA) (Table 1) [21] for 3 weeks to induce vitamin D deficiency. After 3 weeks, blood was collected and analyzed for serum 25-hydroxyvitamin D to confirm vitamin D deficiency. They were then randomly divided into six groups consisting of six representatives as described below:
GP-1: Fed on a VDDD.GP-2: Fed on a VDDD + vitamin D_2_ enriched shiitake mushroom.GP-3: Fed on a VDDD + vitamin D_2_ enriched button mushroom.GP-4: Fed on a VDDD + vitamin D_2_ enriched oyster mushroom.GP-5: Fed on a VDDD + vitamin D_2_ supplement.GP-6: Fed on a VDDD + vitamin D_3_ supplement.

Group 1 (GP-1) was maintained on a vitamin-D-deficient diet throughout the feeding period of 4 weeks and served as the control group of the experiment. The UV-irradiated mushroom (shiitake, button and oyster) powders were weighed to an amount that would contain/supply an amount of 30 IUs/day of vitamin D. Likewise, vitamin D_2_ and D_3_ supplements taken were of an amount that contained 30 IUs of vitamin D. For Wistar rats, the recommended daily intake of vitamin D is estimated to be 1 IU/g of diet [22]. Average diet consumption by Wistar rats (100–150 g) is approx. 20 g/day [23]. Therefore, 30 IUs per day of vitamin D was carefully selected, which reflects about 1.5 IU/g, to simulate vitamin D action. The test diets (mushroom and vitamin D) were suspended in 1 mL deionized water and administered orally for 4 weeks, while all the groups were given free access to deionized water and the vitamin-D-deficient diet with close observation of the animals during the experimental period. Body weight and serum 25-OHD levels were measured on the initial day (0 week) and at 1st week, 2nd week, 3rd week and 4th week of the experiment. Calcium, phosphorous, alkaline phosphatase (ALP) and parathyroid hormone (PTH) levels were checked initially (0 week) and at the end of experiment (after 4 weeks).

### 2.4. Blood Sampling

Blood was collected for biochemical analysis through the retro-orbital plexus under mild ether anesthesia. At the end of the study, rats were given deep anesthesia using 3–5% isoflurane, and blood (3–4 mL/rat) was collected through cardiac puncture, followed immediately by cervical dislocation as an appropriate and humane method of euthanasia. Blood samples were collected in yellow-toped tubes (Nexamo Vacutainer, Mohali, India) and then centrifuged at 1400× *g* for 10 min. The separated serum supernatant was collected and stored at −80 °C for analysis. Repeated thawing and freezing of samples was avoided.

### 2.5. Analysis of Biochemical Parameters

(a).*Serum 25-hydroxy vitamin D:* For the estimation of total vitamin D, the Advia Centaur (Siemens Healthcare Diagnostics Inc., Tarrytown, NY, USA) assay that utilizes a releasing reagent and a monoclonal antibody to detect both 25-OH vitamin D_2_ and D_3_ was used_,_ following standard protocol [24].(b).*Parathyroid hormone levels:* For PTH measurement, Advia Centaur (Siemens Healthcare Diagnostics Inc., Tarrytown, NY, USA) intact PTH assay (monoclonal antibodies assay) was used by following a standard protocol [25].(c).*Serum calcium, phosphorous and alkaline phosphatase:* The estimation was performed by semi-auto-analyzer Erba Chem-5 (Erba Diagnostics, Manheim, Germany) using commercially available kits by following the manufacturer’s protocol [26].

### 2.6. Histology of Femur Samples of Treated and Non-Treated Rats

At the completion of the study the animals were sacrificed and the femur bone was extracted and used to study the bone structure. The bone samples were fixed in cyanuric chloride (0.5%) in methanol containing 1% N-methyl morpholine (0.1 M) for two days [27], then decalcified using formic acid (10%) and stained with hematoxylin and eosin. A digital compound microscope (Motic AE2000, Motic Asia, Kowloon, Hong Kong) was used to examine the slides and histopathological changes; the structure and morphology of the trabecular bone were especially assessed. Histomorphometry variables were analyzed using an image analyzing computer software (Motic images plus 2.0) linked to a microscope [28].

### 2.7. RNA Isolation and Real-Time Quantitative PCR (qPCR) Gene Expression

Immediately after the animal sacrifice, their livers and kidneys were extracted for gene expression analysis. Tissue samples were crushed under liquid nitrogen using an RNAase-free mortar and a pestle [29], and cytoplasmic RNA was extracted using the Q-extra isolation kit (Q-Line Biotech, New Delhi, India) according to the manufacturer’s instructions. The sequence of primers is given in Table 2. The quality and quantity of the extracted RNA were measured using the NanoDrop spectrophotometer (Thermo Fisher Scientific, Madison, WI, USA). The cDNA was synthesized by iScript cDNA synthesis kit (Bio-Rad, USA). Quantitative real-time PCR was performed using iTaq Universal SYBR Green supermix (Bio-Rad, USA) on the QuantStudio^TM^ 5 RTPCR instrument (Thermo Fisher Scientific, Waltham, USA) in triplicate. At a given threshold, Ct value was measured and normalized by the housekeeping gene. The gene expression changes were calculated by the 2^−∆∆CT^ method [30].

### 2.8. Statistical Analysis

All the experimental data were expressed as mean ± standard deviation. Statistical analysis of experimental data for any differences between the groups was performed by analysis of variance (ANOVA) followed by Tukey’s post hoc test. The significance level was set at a *p*-value < 0.05. For the findings of biochemical parameters, a *t*-test was used for statistical analysis to confirm before and after changes in PTH, calcium, phosphorous and ALP levels. GraphPad Prism Instat version 8.0 (GraphPad, San Diego, CA, USA) was used for statistical analyses and for generating all graphs.

## 3. Results

### 3.1. Body Weight

The average body weights of rats during the 4-week feeding period for six groups with different diet plans are shown in Figure 2. No evident effect on the body weight was observed in growing rats fed on different diets, and their mean body weights were not significantly different at any point during the 4-week feeding period. These findings indicated that there was no effect on growth and there were no mortalities within the study period.

### 3.2. Changes in Serum 25-Hydroxyvitamin D Levels

Levels of 25-hydroxy vitamin D in serum of all groups are presented in Table 3. After 4 weeks of the study period, the changes in 25-OHD levels in comparison to the initial measured value for the control group (GP-1) were non-significant (14.48 ± 2.12 to 16.14 ± 3.31 ng/mL, *p* > 0.05), while as for treated groups, GP-2 (11.68 ± 1.92 to 46.00 ± 7.61 ng/mL), GP-3 (16.92 ± 0.48 to 49.96 ± 5.42 ng/mL), GP-4 (15.32 ± 1.28 to 43.62 ± 5.83 ng/mL), GP-5 (12.48 ± 2.12 to 55.14 ± 6.60 ng/mL) and GP-6 (14.22 ± 2.49 to 66.14 ± 6.32 ng/mL), the changes were significant (*p* < 0.05)

### 3.3. Changes in Parathyroid Hormone, Calcium and Phosphorous Levels and Alkaline Phosphatase Activity among All Groups

The biochemical parameters in all groups were measured before and after the feeding period and are stated in Table 4. A significant increase (*p* < 0.05) was found in the levels of calcium and phosphorous, while a significant reduction in alkaline phosphatase and parathyroid hormone readings was observed in all the treated groups after 4 weeks of the feeding period. Non-significant (*p* > 0.05) changes were observed for all the parameters in the control group.

### 3.4. Bone Health of Rats Fed with Vitamin-D-Deficient Diet, Edible Mushrooms and Vitamin D Supplements

Histopathological changes between the control group (fed only on a customized vitamin-D-deficient diet) and treated groups (rat group fed additionally with shiitake, button and oyster mushrooms and vitamin D_2_ and D_3_ supplements) are shown in Figure 3. No significant histopathological differences were observed in the grossly cut sections. However, moderate osteoporotic changes were revealed during microscopic examination showing wide and separated bone trabeculae, and there was also a reduction in the thickness of the bone cortex in GP-1, while benign bone tissue and trabeculae were observed in the bone structures of mushroom-fed (GP-2, GP-3 and GP-4) and vitamin-D-supplemented (GP-5 and GP-6) groups. The findings (Figure 4) revealed a significant increase in trabecular separation and a significant reduction in the osteoid area in the selected region of interest with respect to the control group (*p* < 0.05).

Figure 3 (GP-2 to GP-4) shows structural changes in the femur bone of vitamin-D-deficient rats after feeding on vitamin-D-enriched shiitake, button and oyster mushrooms for 4 weeks. Trabecular width in all three mushroom-fed groups was improved together with a decline in bone marrow spaces. The measured dimensions of the osteoid area and trabecular separation in mushroom-fed groups (Figure 4) revealed a significant increase in the osteoid area (shiitake: 19 ± 0.9; button: 22 ± 1.1; oyster:18 ± 1.2 vs. control group 11.10 ± 0.9 mm^2^ with *p* < 0.05) and a significant decrease in trabecular separation (shiitake: 23 ± 0.8; button 22 ± 0.6; oyster 24 ± 1.1 vs. control 31.10 ± 2.2 mm^2^ with *p* < 0.05).

Vitamin D_2_ and D_3_ had a prevalent effect on the femur bone structure and microscopic examination showed a significant improvement, while the histomorphometry measurements (Figure 3, GP-5 and GP-6) after 4 weeks of treatment revealed a significant increase in the osteoid area (D_2:_ 22 ±1.0 and D_3_ 24.1 ± 2.1 vs. control 11.10 ± 0.9 mm^2^ with *p* < 0.05) and a decrease in the trabecular separation (D_2:_ 22 ± 0.6 and D_3_ 22 ± 0.5 vs. control 31.10 ± 2.2 mm^2^ with *p* < 0.05).

### 3.5. Expression of CYP2R1, CYP27B1 and VDR Gene in the Liver and Kidney of Rats after 4 Weeks of Feeding Period

After the qPCR assay, we noticed high expression of VDR and CYP2R1 genes in liver for mushroom-fed and vitamin-D-supplemented groups. As shown in Figure 5 and amplification plots (Appendix A), it was observed that in the group fed with button mushrooms CYP2R1 was upregulated (Ct = 27.56). The vitamin D receptor was significantly upregulated in the button mushroom (2.47 folds) and vitamin D_3_ (2.5 folds) groups. Moreover, we observed downregulation of CYP27B1 in the liver. In the kidney, CYP2R1 expression was very weak. In response to mushroom feeding as well as vitamin D_2_/D_3_ supplementation, a significant upregulation in VDR mRNA expression was observed; however, in vitamin-D_2_- and D_3_- supplemented groups the expression was relatively higher (~50-fold and ~24-fold, respectively). Further, CYP27B1 expression in the kidney was approximately 1.8-, 6.8- and 1.5-fold higher in the shiitake-, button- and oyster-fed groups (GP-2, GP-3 and GP-4), respectively, with respect to the control group (GP-1). Further, a non-significant increase in CYP27B1 expression in the vitamin-D_2_-supplemented group (GP-5) was observed, while a downregulation in the vitamin D_3_ group (GP-6) was observed (Figure 5).

## 4. Discussion

Vitamin D deficiency has turned out to be a major metabolic disorder that primarily affects bone health by retarding bone growth and development and results in the symptoms of rickets and osteoporosis [31]. Vitamin-D-deficient rat models are frequently used in vitamin D research. Deprivation of dietary vitamin D and sunlight is a traditional technique used to produce vitamin-D-deficient animal models [32]. To ascertain vitamin D deficiency, low levels of 25-OHD as well as increased PTH levels (secondary hyperparathyroidism) and alkaline phosphatase activity in blood were attained. The similar trend of results with decreased 25-OHD levels, raised PTH levels and ALP activity in vitamin-D-deficient rats have been reported by earlier studies [28,33]. Parathyroid hormone is important for bone health by maintaining the normal concentrations of serum calcium and phosphorous. With the increase in PTH levels, there is an increase in bone turnover, thus causing mineral imbalances in bones leading to increased fracture risk [34]. A symptom common of rickets and osteomalacia is secondary hyperparathyroidism, which in turn is a result of vitamin D deficiency [35]. Additionally, ALP also plays its part in phosphorus homeostasis and bone mineralization [36]. It has been reported to increase in conditions of osteomalacia and rickets [37,38], though it cannot be considered a reliable biomarker of vitamin D deficiency [39].

In the present study, Wistar rats were fed on a customized vitamin-D-deficient diet to induce hypovitaminosis D. All vitamin-D-deficient groups (GP-1, GP-2, GP-3, GP-4, GP-5 and GP-6) showed variable degrees of secondary hyperparathyroidism (PTH > 65 pg/mL cutoff value, [40]), normal calcium (range 5.3–13 mg/dL [41]), low serum phosphorus levels (range 2.7–4.5 mg/dL [42]) and increased alkaline phosphatase levels (range 44 to 147 IU/L [43]).

Further, histopathological and histomorphometric measurements revealed moderate osteoporotic changes with a significant reduction in bone area and a wide separation of bone trabeculae. The findings of the present study were completely in line with a previous study by Abulmeaty [28], who reported a similar trend of biochemical changes in the blood parameters as well as in the bone histomorphometry of the vitamin-D-deficient rats. The findings were also in line with another previous study of Kollenkirchen et al. [44], with a difference that in their study, the vitamin-D-deficient rat model in addition to normal calcium levels had normal phosphate and PTH levels. That was maybe due to the inclusion of dietary lactose, which promoted passive absorption of calcium in the intestine [45]. Additionally, the biochemical and histological changes in vitamin-D-deficient rats were also analogous to another previous study [46].

Even though 25-hydroxy vitamin D is considered as the marker of vitamin D in the body and current guidelines of most of endocrine societies endorse its assessment for vitamin D deficiency screening, the non-assessment of 1,25-dihydroxy vitamin D levels could be considered as a limitation of this study. However, serum 1,25-dihydroxy vitamin D levels have little or no impact on the vitamin D stores in the body and are regulated mainly by parathyroid hormone, which in turn is regulated by calcium and/or vitamin D [47].

Initially, when all the groups were fed with the vitamin-D-deficient diet, hyperparathyroidism was a common finding. Active vitamin D metabolites decline PTH production [48]. Some previous studies suggest that 1,25-dihydroxy vitamin D, irrespective of the changes in intestinal calcium absorption and serum calcium, can limit the transcription of PTH by binding to the vitamin D receptor which forms a heterodimer with the retinoid X receptor that binds to the vitamin D response elements within the PTH gene, regulating its transcription [49]. In addition, PTH secretion is indirectly altered by calcium-sensing receptors whose expression is also regulated by 1,25-dihydroxy vitamin D. Thus, reduced concentrations of vitamin D and calcium-sensing receptors increase PTH secretion [50]. The above facts could be a possible explanation for hyperparathyroidism in vitamin-D-deficient rats in our study. Vitamin D deficiency caused a condition of hyperparathyroidism despite having normal calcium levels, demonstrating a relatively narrow range of PTH secretion regulation by extracellular calcium [51]. The significant improvement in the PTH levels of mushroom-fed (GP-2, GP-3 and GP-4) and vitamin-D-supplemented (GP-5 and GP-6) groups is primarily due to the vitamin-D-induced increase in vitamin D active metabolites and calcium-sensing receptors.

The significant increase in 25-OHD level in mushroom-fed groups is a sign of good bioavailability of vitamin D and help in restoring vitamin D status and bone structure. These explanations are backed by significant before and after changes in histological, histomorphometry and biochemical parameters. Furthermore, the findings regarding vitamin D supplementation were in line with a previously conducted study [28] where the vitamin-D_3_-fed group had shown a significant increase in 25-OHD levels. The changes in our study were also more prominent in the vitamin-D_3_-supplemented group. In addition, Calvo et al. [18] reported that diets containing UV-light-exposed mushrooms significantly increased circulating serum 25-OHD levels and repressed PTH levels in comparison to rats fed on mushrooms unexposed to UV light.

The mean changes in 25-hydroxyvitamin D levels at week 0 and after 4 weeks of study were 34.32 ng/mL for the shiitake mushroom, 33.04 ng/mL for the button and 28.30 ng/mL for the oyster mushroom, and for vitamin D_2_ and vitamin D_3_ the values were 42.66 ng/mL and 51.92 ng/mL, respectively. The changes were statistically significant (*p* < 0.05), while non-significant changes were observed in the control group (1.66 ng/mL). These data were consistent with the results of previous studies [16,18], who have reported that UV-irradiated button mushrooms were effective in increasing serum 25-OHD levels. A large difference in the response to mushroom feeding and vitamin D supplementation can also be observed in contrast to the PTH levels of vitamin-D-deficient rats. Moreover, the histological changes were more evident in the vitamin D_3_ group, which demonstrates that vitamin D_3_ is slightly more effective than vitamin D_2_ or mushroom feed in improving vitamin D status and its related conditions.

There has been enough research determining the skeletal effects of vitamin D; however, non-skeletal effects remain controversial. Feeding vitamin-D-enriched mushrooms or vitamin D_2_ and D_3_ to vitamin-D-deficient groups and investigating the changes in the gene expression will allow to predict the response of a specific organ and its functionality. In the present study, we observed that the mushrooms, as well as vitamin D supplementation, did change the expression of the CYP2R1 and VDR genes in the liver (Figure 5), although there is contradictory information on the regulation of the vitamin-D-related genes by vitamin D supplementation. Recently, researchers have reported that CYP27B1 and VDR expression is independent of serum vitamin D metabolite levels [52]; however, another earlier study suggests that 1,25-dihydroxy vitamin D levels increased VDR gene expression [53]. Moreover, it was suggested that in rats, 1,25-dihydroxy vitamin D downregulates VDR and upregulates CYP27B1 [54]. Another study confirms that 1,25-dihydroxy vitamin D suppresses the renal CYP27B1 [55]. We also observed apparent downregulation of the CYP27B1 gene by vitamin D_3_ supplementation; however, there was an upregulation in groups fed with vitamin-D-rich mushrooms. Furthermore, in the present study, we did not measure 1,25-dihydroxy vitamin D levels, which may be considered as a drawback. However, in line with our results, a previous study reported increased VDR and CYP2R1 expression in the liver and increased VDR and CYP27B1 in renal tissue in response to vitamin D supplementation [56]. To the best of the authors’ knowledge, no previous study has reported the effect of vitamin-D-enriched mushrooms on the expression of VDR, CYP2R1 and CYP27B1 genes, and thus, this encourages further investigation.

From the findings it was observed that all three mushrooms led to an increase in 25-OHD levels; however, there are certain differences in calcium, phosphorus and ALP levels as well as in the expressions of CYP isoenzymes and VDR in the liver and the kidney. The exact reason for this difference is not known. However, the possible variation may arise due to species differences. In addition, the levels of PTH, calcium, phosphorous and ALP are mainly linked to the active metabolite of vitamin D, i.e., 1,25-dihydroxy vitamin D; however, 1,25-dihydroxy vitamin D was not analyzed in this study, and this can be considered as a limitation.

Considering the worldwide deficiency of vitamin D, the use of UV irradiation to increase the vitamin D_2_ content in mushrooms could prove as an effective strategy to improve 25-OHD levels. An earlier study by Simon et al. [57] to investigate the suitability of UV light for vitamin D enrichment in mushrooms found that the effects of UVB light are only limited to vitamin D formation. and no such detrimental changes on the nutritional composition were identified. Further, the UVB exposure process for the production of button mushrooms has been approved in European Union under novel food regulation, i.e., regulation (EU) 2015/2283 [58]. In addition, other studies have reported that the UV exposure has a negative effect on the color of edible mushrooms; however, in our recent study [20], we observed that there was no significant effect on the color profile of edible mushrooms exposed to UVB for 120 min.

Furthermore, in a recent study [59] it was found that vitamin D_2_ is stable in the irradiated mushrooms during storage. Therefore, ultraviolet-exposed mushrooms can be considered as a reliable dietary source of vitamin D_2._ The present study, which mainly focused on commercial species, viz., button, shiitake and oyster mushrooms, revealed that among the three varieties, the button mushroom is the most promising in terms of cost, vitamin D_2_ yield, as well as the pharmacological results. It should be noted that vitamin D_3_ is not preferred by many vegetarians, and thus vitamin D_2_ from edible mushrooms is a suitable alternative.

## 5. Conclusions

In conclusion, the role of vitamin-D_2_-enriched mushrooms may extend beyond the stabilization of serum 25-OHD levels. Vitamin D_2_ from UVB-irradiated mushrooms is bioavailable and functional in maintaining bone health and mineralization in rat models without any apparent adverse effect. Our results demonstrated that the VDR was strongly expressed in both the liver and kidney, while CYP2R1 is more expressed in the liver and CYP27B1 is prevalent in the kidney. Future studies are needed for thorough toxicological examination to confirm that UV-irradiated mushrooms are safe for human consumption.

## Figures and Tables

**Figure 1 jof-08-00864-f001:**
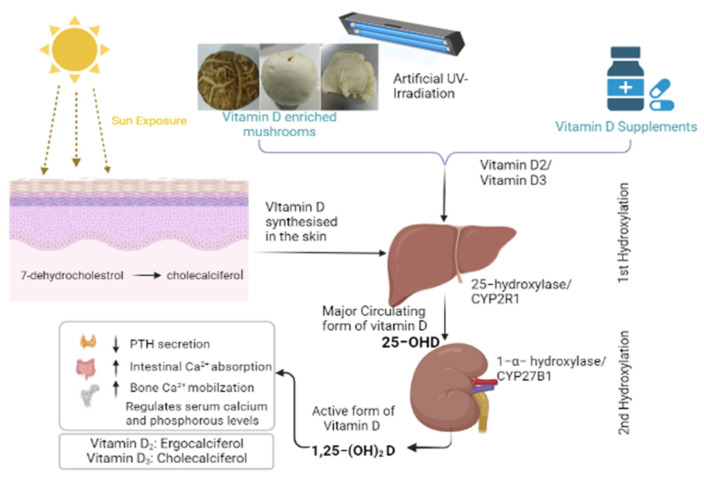
Metabolism of vitamin D in the body.

**Figure 2 jof-08-00864-f002:**
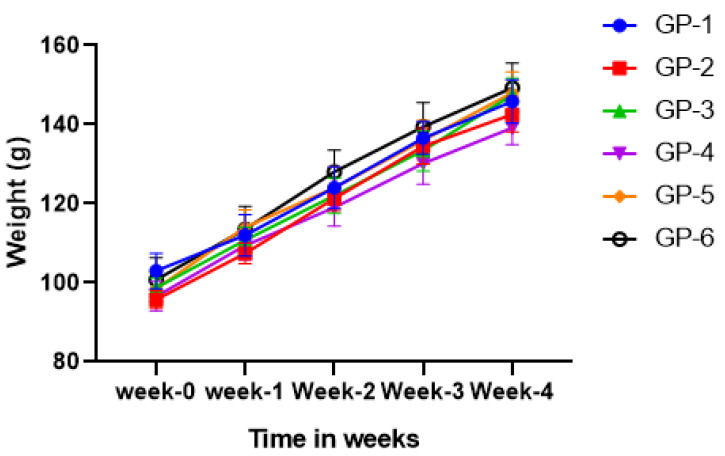
Effect of UV-exposed mushrooms and vitamin D on the growth of Wistar rats. There were no statistically significant differences between groups at any time. Each point represents the mean ± SD for n = 6.

**Figure 3 jof-08-00864-f003:**
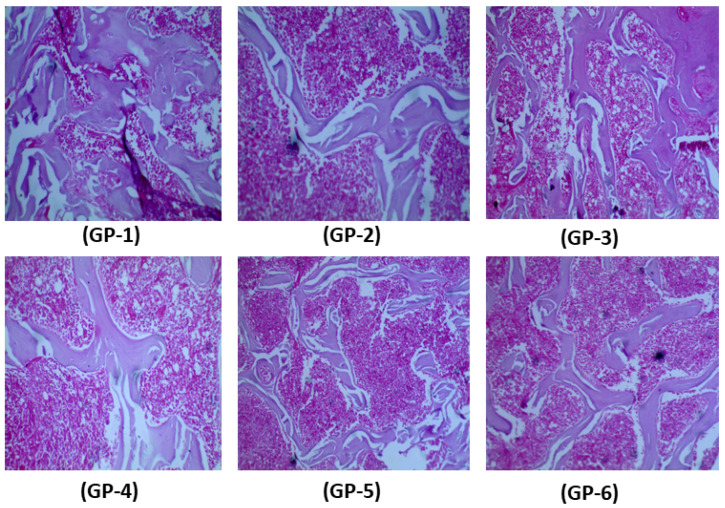
Histopathological changes in bone among group fed on vitamin D deficient diet (GP-1), shiitake (GP-2), button (GP-3), oyster (GP-4), vitamin D_2_ (GP-5) and vitamin D_3_ (GP-6).

**Figure 4 jof-08-00864-f004:**
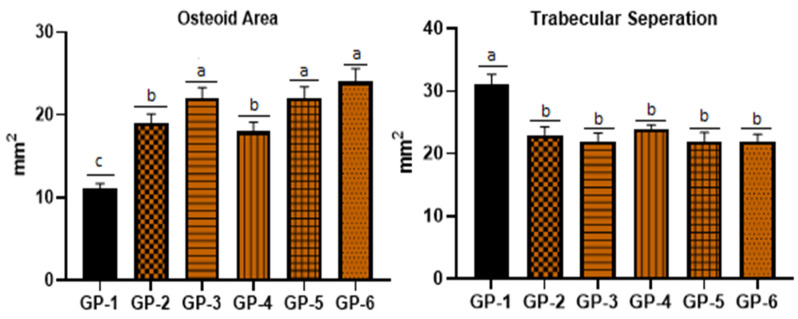
Histomorphometric difference in osteoid area and trabecular separation among group fed on vitamin D deficient (GP-1), shiitake (GP-2), button (GP-3), oyster (GP-4), vitamin D_2_ (GP-5) and vitamin D_3_ (GP-6). Values are presented as mean ± SD for n = 6. The experimental values within any specific group that do not have a common superscript are significantly different (*p* < 0.05) based on Tukey’s post hoc test.

**Figure 5 jof-08-00864-f005:**
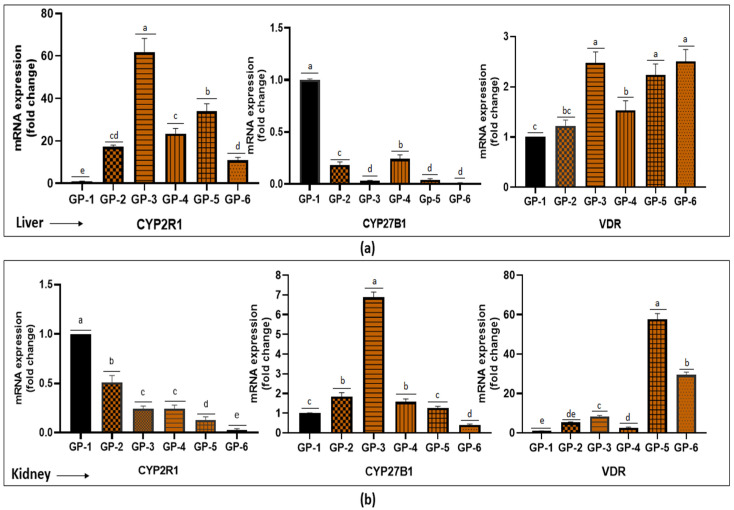
Effect of UV-exposed mushrooms and vitamin D supplementation on the expression of CYP2R1, CYP27B1 and VDR genes in the liver (**a**) and kidney (**b**). Values are presented as mean ± SD for n = 6. The experimental values within any specific group that do not have a common superscript are significantly different (*p* < 0.05) based on Tukey’s post hoc test.

**Table 1 jof-08-00864-t001:** Vitamin-D-deficient diet based on AIN-93G composition.

Ingredient	Concentration
Carbohydrates	60%
Fat	7%
Protein	18%
Fiber	5%
Ash	2.5%
Calcium	5 g/Kg
Phosphorous	2.5 g/Kg
Vitamin D	0.00 IU
Total energy	3.85 Kcal/g

**Table 2 jof-08-00864-t002:** List of primers and their sequences used for real-time PCR analysis.

Primer	Symbol	Sequence 5′ → 3′
Vitamin D receptor	VDR	F-TGTTCACCTGTCCCTTCAAT R-CGCTGTACCTCCTCATCTGT
Cytochrome P450 2R1	CYP2R1	F-CCTTCT’GCTACTACTCGTGC R-GCATGGTCTATCTC’GCCAAA
Cytochrome P450 27B1	CYP27B1	F-TTTCTCATCTTGGTCAGAGC R-AGAGTGTAGACACAAACACC
Glyceraldehyde 3-phosphate dehydrogenase	GAPDH	F-GGGTGTGAACCACGAGAAATA R-AGTTGTCATGGATGACCTTGG

**Table 3 jof-08-00864-t003:** Serum 25-hydroxyvitamin D levels (ng/mL) of all groups measured at weekly intervals.

Groups	0th Week	1st Week	2nd Week	3rd Week	4th Week
GP-1	14.48 ± 2.12 ^e^	14.69 ± 1.18 ^e^	11.48 ± 2.12 ^e^	13.42 ± 1.42 ^e^	16.14 ± 3.31 ^e^
GP-2	11.68 ± 1.92 ^e^	18.93 ± 3.31 ^d^	31.68 ± 4.92 ^c^	37.17 ± 4.62 ^c^	46.00 ± 7.61 ^b^
GP-3	16.92 ± 0.48 ^e^	24.56 ± 2.73 ^d^	36.92 ± 5.48 ^c^	41.64 ± 8.83 ^b^	49.96 ± 5.42 ^b^
GP-4	15.32 ± 1.28 ^e^	22.05 ± 4.67 ^e^	35.32 ± 3.28 ^c^	39.33 ± 3.32 ^c^	43.62 ± 5.83 ^b^
GP-5	12.48 ± 2.12 ^e^	21.69 ± 3.15 ^d^	33.48 ± 4.12 ^c^	46.42 ± 6.20 ^b^	55.14 ± 6.60 ^ab^
GP-6	14.22 ± 2.49 ^e^	28.17 ± 4.28 ^d^	37.48 ± 2.08 ^c^	55.53 ± 5.42 ^ab^	66.14 ± 6.32 ^a^

Data were expressed as mean ± SD where n = 6. One-way ANOVA with Tukey’s post hoc test was used to test the significant difference. The experimental values that do not have a common superscript are significantly different (*p* < 0.05).

**Table 4 jof-08-00864-t004:** Changes in the biochemical parameters before and after 4 weeks of feeding period.

Groups	PTH (pg/mL)		Calcium (mg/dL)		Phosphorous (mg/dL)		ALP (U/L)	
	0th Week	4th Week	0th Week	4th Week	0th Week	4th Week	0th Week	4th Week
GP-1	66.69 ± 4.18 ^a^	69.11 ± 6.11 ^a^	5.48 ± 2.12 ^a^	6.78 ± 1.18 ^a^	1.21 ± 0.12 ^a^	1.42 ± 0.18 ^a^	164.14 ± 27.31 ^b^	157 ± 23.13 ^b^
GP-2	78.93 ± 5.31 ^a^	28.85 ± 3.18 ^b^	5.28 ± 1.92 ^b^	9.93 ± 1.21 ^a^	1.17 ± 0.22 ^b^	3.16 ± 0.78 ^a^	171.00 ± 36.61 ^b^	82 ± 13.25 ^a^
GP-3	74.05 ± 4.67 ^a^	16.87 ± 2.12 ^b^	5.32 ± 1.28 ^b^	10.70 ± 1.48 ^a^	1.32 ± 0.32 ^b^	3.34 ± 1.11 ^a^	182.62 ± 41.83 ^b^	79 ± 16.18 ^a^
GP-4	65.56 ± 7.73 ^a^	28.10 ± 4.23 ^b^	5.72 ± 0.48 ^b^	8.50 ± 0.78 ^a^	1.46 ± 0.23 ^b^	2.85 ± 0.88 ^a^	175.96 ± 35.42 ^b^	65 ± 11.09 ^a^
GP-5	67.84 ± 6.16 ^a^	18.94 ± 3.88 ^b^	6.29 ± 1.15 ^b^	10.23 ± 2.01 ^a^	2.11 ± 0.34 ^b^	3.27 ± 1.18 ^a^	158.50 ± 14.47 ^b^	61 ± 15.33 ^a^
GP-6	72.34 ± 5.98 ^a^	15.25 ± 5.10 ^b^	6.04 ± 0.92 ^b^	10.75 ± 1.85 ^a^	1.97 ± 0.47 ^b^	3.78 ± 1.22 ^a^	168.30 ± 18.17 ^b^	58 ± 13.18 ^a^

Values are expressed as mean ± SD for triplicate experiments (n = 6). Values that do not share a common superscript are significantly different at *p* < 0.05.

## Data Availability

The data presented in this study are available on request from the corresponding authors.

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
