# Peer review of "Effect of Vitamin-D-Enriched Edible Mushrooms on Vitamin D Status, Bone Health and Expression of CYP2R1, CYP27B1 and VDR Gene in Wistar Rats"

_jof, 2022, doi:10.3390/jof8080864_

Round 1
Reviewer 1 Report
Effect of Vitamin D Enriched Edible Mushrooms on Vitamin D Status, Bone Health and Expression of CYP2R1, CYP27B1 and VDR Gene in Wistar Rats
The authors took vitamin D deficient rats and feed the three commercially available mushrooms (UV to increase Vitamin D production) in addition to vitamin D supplements. The information presented has interest as increasing vitamin D in the diet is important for health. I did enjoy reading this manuscript but there are some issues that need to be addressed, see below.
Methodology issue: Unless I missed it, the authors did not examine the levels of ergocalciferol/vitamin D2 before and after UV irradiation in this study. I understand there is research stating this occurs but without verifying this for this experiment, I don’t understand how the authors can state vitamin D enriched (there must be verification). There is always a possibility that the UV enrichment treatment did not work this time, which could mean something else caused the results they are recording. Please add this into the manuscript. The authors also need to address why there was no non-enrichment diet as a negative control. Fungi produce a lot of compounds and if the enrichment procedure is being evaluated, a non-enrichment diet control is more appropriate.
Additional issues:
Abstract: Line 24: feeding period in…: Sentence is awkward as written, please revise.
Keywords: All words in the title are key words, therefore, please consider replacing duplicated key words.
Introduction: Many of the sentences need to be reviewed as they are awkward as written. Some sentences should be broken into two sentences. Examples below.
Line 38: Should not start a sentence with a number. Either revise the sentence or spell out the first word of a sentence.
Line 40: Consider revising to: “…more than 50% are vitamin D deficient [5].”
Line 40-42: This sentence is awkward as written: Consider revising to something like: Some Middle East countries like the United Arab Emirates and Saudi Arabia also report high levels of vitamin D deficiency (50% and 59% respectively) despite having abundant sunshine.
Line 45-47: This sentence is awkward as written: Consider revising to something like: Dietary sources of vitamin D3 include animal sources, while dietary sources of vitamin D2 are phytoplankton, invertebrates, and Fungi [8].
Line 67: sp. should not be italicized
Line 72: change restorative to beneficial
Restorative is too vague
Material and Methods: needs to be standalone, needs more detail in places.
2.1: The authors site their manuscript which is fine, but this manuscript must be standalone in its methods. Please add a few statements about the procedure needed for vitamin D enrichment. Also, please consistently use enrichment (replace the work enhancement with enrichment). Also, if you measured the before and after to show this treatment worked (control), the authors need to add the methodology used to test the vitamin D concentrations.
Line 141: … and after (4th week) feeding period. This is awkward as written, please revise.
Line 177: revise to: The sequence primers are given in…
Line 192-193: revise to: GraphPad Prism Instat version 8.0(GraphPad, San Diego, CA, USA) was used for statistical analyses and for generating all graphs.
Results: This is a standalone results section (Not a Results and discussion section) so there can be no interpretation (example line 199-201) within this section. Clearly state the results, that is all. Also, please consult the instructions to authors for the figure and table captions. The authors have these in different styles (example line 197, 271 and 250). Also, Table is capitalized while figure is not in the manuscript. Please review the instructions to authors for clarification on style.
Table 3. (Line 218): it looks like there are superscript for both column and rows? If so, please update this table footer.
Line 241-242: Tools of Motic…: This sentence is awkward as written but this is method and should be removed.
Line 248-259: There are words in these sentences that should be removed because they do not clearly state the results. For example: enhanced, precise measured, improvement. These sentences are also awkward as written. Just state the results clearly as they are. Enhanced and improvement are vague and subjective. No need for use precise measured, if not measure precisely then the data should not be reported so it is no needed. These only make it harder to read.
Line 272: was proactive in …: Proactive? How was this measured? It’s too vague. Please remove, just say it was upregulated.
Line 273: VDR: Please spell out the first work of a sentence.
Discussion: The discussion is too vague. The authors state in several places that their research is similar to others (examples: Line 358) but never say what the other authors found. Are they similar in that there are elevated levels or are the blood levels similar? Please let the reader know how they are similar, it's not enough to just as our results were similar to [100]. Explain how they are similar
Conclusion: 392: The last sentence ends vague: Future studies are needed for more precise elucidation about this. Define this please.
Author Response
Response to Reviewer 1
The authors took vitamin D deficient rats and feed the three commercially available mushrooms (UV to increase Vitamin D production) in addition to vitamin D supplements. The information presented has interest as increasing vitamin D in the diet is important for health. I did enjoy reading this manuscript but there are some issues that need to be addressed, see below.
Methodology issue: Unless I missed it, the authors did not examine the levels of ergocalciferol/vitamin D2 before and after UV irradiation in this study. I understand there is research stating this occurs but without verifying this for this experiment, I don’t understand how the authors can state vitamin D enriched (there must be verification). There is always a possibility that the UV enrichment treatment did not work this time, which could mean something else caused the results they are recording. Please add this into the manuscript. The authors also need to address why there was no non-enrichment diet as a negative control. Fungi produce a lot of compounds and if the enrichment procedure is being evaluated, a non-enrichment diet control is more appropriate.
Response: We appreciate the reviewers’ comments, which helped a lot to improve the manuscript. As suggested, we have included the procedure of irradiation for vitamin D2 enrichment. We want to share with the reviewer that we analysed/verified the vitamin D2 content in all the three mushroom varieties. The amount of vitamin D2 increased form negligible (1.9 ± 0.17, 1.1 ± 0.11, 2.2 ± 0.24) to considerable (25.6 ± 0.42, 32.4 ± 1.45 and 28.7 ± 1.14 μg/g respectively) levels after irradiation. we have added this into the revised manuscript (L 119-144). The focus of our study was to evaluate the effect of vitamin D2 from enriched mushrooms on vitamin D status and it was clear from the initial studies that non enriched mushrooms were lacking vitamin D, so it was assumed that non enriched group will not produce the desired effect. This is the reason for not considering non enriched mushroom diet.
Additional issues:
Abstract: Line 24: feeding period in…: Sentence is awkward as written, please revise.
Response: we have rewritten the sentence (L 26-27)
Keywords: All words in the title are key words, therefore, please consider replacing duplicated key words.
Response: We have replaced the duplicate key words (L 35-36)
Introduction: Many of the sentences need to be reviewed as they are awkward as written. Some sentences should be broken into two sentences. Examples below.
Line 38: Should not start a sentence with a number. Either revise the sentence or spell out the first word of a sentence.
Response: As correctly suggested we have revised the sentence (L 43-44)
Line 40: Consider revising to: “…more than 50% are vitamin D deficient [5].”
Response: We have revised as suggested (L 45)
Line 40-42: This sentence is awkward as written: Consider revising to something like: Some Middle East countries like the United Arab Emirates and Saudi Arabia also report high levels of vitamin D deficiency (50% and 59% respectively) despite having abundant sunshine.
Response: We are thankful to the reviewer for this important suggestion, we have revised the sentence as suggested (L 46-48)
Line 45-47: This sentence is awkward as written: Consider revising to something like: Dietary sources of vitamin D3 include animal sources, while dietary sources of vitamin D2 are phytoplankton, invertebrates, and Fungi [8].
Response: We have revised the sentence as suggested (L 53-55)
Line 67: sp. should not be italicized
Response: Word has been corrected (L 75)
Line 72: change restorative to beneficial
Restorative is too vague
Response: Word has been changed as suggested (L 81)
Material and Methods: needs to be standalone, needs more detail in places.
2.1: The authors site their manuscript which is fine, but this manuscript must be standalone in its methods. Please add a few statements about the procedure needed for vitamin D enrichment. Also, please consistently use enrichment (replace the work enhancement with enrichment). Also, if you measured the before and after to show this treatment worked (control), the authors need to add the methodology used to test the vitamin D concentrations.
Response: We are thankful to the reviewer for this valuable suggestion and providing an opportunity to improve the manuscript. We have added the procedure for vitamin D enrichment in the revised manuscript (L119-126). We have replaced the word “enhancement’ as suggested (L 68, 72). We want to further share with the reviewer that we did analysed vitamin D content using HPLC, method included in the text (L 127-143). The concentration of vitamin D2 in shiitake, button and oyster mushroom increased from negligible to 25.6 ± 0.42, 32.4 ± 1.45 and 28.7 ± 1.14 μg/g respectively (L142-144).
Line 141: … and after (4th week) feeding period. This is awkward as written, please revise.
Response: We have revised the sentence as suggested (L 181)
Line 177: revise to: The sequence primers are given in…
Response: We have incorporated the changes as suggested (L 219)
Line 192-193: revise to: GraphPad Prism Instat version 8.0(GraphPad, San Diego, CA, USA) was used for statistical analyses and for generating all graphs.
Response: We have revised the sentence as suggested (L 234-235)
Results: This is a standalone results section (Not a Results and discussion section) so there can be no interpretation (example line 199-201) within this section. Clearly state the results, that is all. Also, please consult the instructions to authors for the figure and table captions. The authors have these in different styles (example line 197, 271 and 250). Also, Table is capitalized while figure is not in the manuscript. Please review the instructions to authors for clarification on style.
Response: We agree with the reviewer and we have removed the line (252-254). Further we have updated all the Figure and Table captions as per journal guideline in the revised manuscript (L 96, 241, 280, 288, 295)
Table 3. (Line 218): it looks like there are superscript for both column and rows? If so, please update this table footer.
Response: We appreciate the reviewer for this important suggestion, we have updated the table footer (L 262)
Line 241-242: Tools of Motic…: This sentence is awkward as written but this is method and should be removed.
Response: We have removed the sentence as suggested (L 286-287)
Line 248-259: There are words in these sentences that should be removed because they do not clearly state the results. For example: enhanced, precise measured, improvement. These sentences are also awkward as written. Just state the results clearly as they are. Enhanced and improvement are vague and subjective. No need for use precise measured, if not measure precisely then the data should not be reported so it is no needed. These only make it harder to read.
Response: We are thankful to the reviewer for these minute but valuable suggestions, we have thoroughly revised the manuscript for the above stated mistakes (L 294, 294, 302).
Line 272: was proactive in …: Proactive? How was this measured? It’s too vague. Please remove, just say it was upregulated.
Response: We have removed the word and has revised the sentence as suggested (L 318)
Line 273: VDR: Please spell out the first work of a sentence.
Response: Sentence has been revised as suggested (L 317)
Discussion: The discussion is too vague. The authors state in several places that their research is similar to others (examples: Line 358) but never say what the other authors found. Are they similar in that there are elevated levels or are the blood levels similar? Please let the reader know how they are similar, it's not enough to just as our results were similar to [100]. Explain how they are similar
Response: We appreciate the reviewer for this important comment, we have explained the findings of other authors and updated the discussion section in the revised manuscript (L 342-343, 361-362, 397-398, 408-409)
Conclusion: 392: The last sentence ends vague: Future studies are needed for more precise elucidation about this. Define this please.
Response: We agree with the reviewer and as per the suggestions of reviewer 3 about toxicological examinations, we have revised the sentence (L 470-471).
Dr. Bibhu Prasad Panda
Corresponding author
Reviewer 2 Report
It is opinion of the reviewer that this paper before acceptance Leeds several corrections/revisions. My individual comments are listed below.
L. 7 – 15 – The authors’ initials and e-mail addresses must be completed.
L. 19 – What does it mean “nutritional foods?
L. 21 – It should be “… of ultraviolet irradiation …”.
In the Introduction section, the bioavailability of vitamin D from plants/mushrooms should be reported.
L. 69 – It should be “.. high content of nutritional …”.
L. 71 – Remove “antioxidants”.
L. 73 – It should be “tocpoherols”.
L. 97 – What does it mean “vitamin D supplemenys”?
L. 101 – The irradiation must be described.
L. 116 it should be “fluorescent”.
L. 150 – The centrifugation must be characterized by “x g” instead of “rpm”.
L. 160 – It should be “… phosphorus and alkaline …”.
L. 208 – It should be “hydroxy”
L. 220- – It should be “… phosphorus level and alkaline phosphatase activity …”.
Table 3, 4, figure 4, 5 – The highest value should be marked with “a”, lower with “b”, etc.
L. 303 – What does it mean “a rat model fed on …”?
L. 315 – What does it mean “ in Kollenkirchen vitamin D …”?
L. 323/324 – “1,25 dihydroxy-vitamin D” , “1,25-dihydroxy vitamin” , – nomenclature needs unification.
L. 118, 295, – “25(OH)D”, “25-hydroxyvitamin D”
L. 318 – It should be “in the intestine”.
L. 385 – It should be “Conclusions”.
L. 393 – It should be initials instead of full names.
References – Big mess! References must be in the MDPI style: abbreviations not full journal names, remove months, add DOI.
Author Response
Response to Reviewer 2
Comments and Suggestions for Authors
It is opinion of the reviewer that this paper before acceptance Leeds several corrections/revisions. My individual comments are listed below.
- 7 – 15 – The authors’ initials and e-mail addresses must be completed.
Response: We appreciate the very valuable suggestion by the reviewer. We have added initials along with e-mail addresses. (L 7-20)
- 19 – What does it mean “nutritional foods?
Response: Though mushrooms are low-calorie, low-fat foods, they are considered as nutritional foods as they are not just proteinaceous but a complete health food providing all the nutrients (high content of fibres, minerals, vitamins, etc.)
- 21 – It should be “… of ultraviolet irradiation …”.
Response: UV is abbreviation for ultraviolet used in first place, we have corrected by keeping UV in brackets (L 24)
In the Introduction section, the bioavailability of vitamin D from plants/mushrooms should be reported.
Response: Thank you for the important suggestions, we have updated the introduction section as suggested. (L 85-89).
- 69 – It should be “.. high content of nutritional …”.
Response: As suggested sentence has been rewritten. (L 76)
- 71 – Remove “antioxidants”.
Response: Agreed, Text has been updated. (L 80)
- 73 – It should be “tocpoherols”.
Response: we have updated (L 81)
- 97 – What does it mean “vitamin D supplemenys”?
Response: Vitamin D supplements are the two available forms of vitamin D (Vitamin D2 and Vitamin D3) (L 115)
- 101 – The irradiation must be described.
Response: We have described the irradiation procedure in the revised manuscript. (L 119-126)
- 116 it should be “fluorescent”.
Response: Updated. (L 155)
- 150 – The centrifugation must be characterized by “x g” instead of “rpm”.
Response: Agreed, “rpm” has been changed to “g” in the revised manuscript. (L 191)
- 160 – It should be “… phosphorus and alkaline …”.
Response: We have updated the text in revised manuscript. (L 201)
- 208 – It should be “hydroxy”
Response: Text has been updated. (L 253)
- 220- – It should be “… phosphorus level and alkaline phosphatase activity …”.
Response: Agreed, Text has been updated. (L 265-266)
Table 3, 4, figure 4, 5 – The highest value should be marked with “a”, lower with “b”, etc.
Response: As suggested, the superscript letters have been remarked with “a” as highest and likewise (Table 3,4, Figure 4,5)
- 303 – What does it mean “a rat model fed on …”?
Response: We have updated the text to remove vagueness (L 352).
- 315 – What does it mean “ in Kollenkirchen vitamin D …”?
Response: It was “….in their study vitamin D….”, we have corrected the text (364).
- 323/324 – “1,25 dihydroxy-vitamin D” , “1,25-dihydroxy vitamin” , – nomenclature needs unification.
Response: 1,25-dihydroxy vitamin, it has been unified throughout the revised manuscript (L 96, 371-372, 378, 383-384, 422, 425, 426-427, 430).
- 118, 295, – “25(OH)D”, “25-hydroxyvitamin D”
Response: we have updated the text for nomenclature unification in the revised manuscript (L 25, 40, 91, 178, 256, 340, 342, 393, 398, 401)
- 318 – It should be “in the intestine”.
Response: Agreed, Text has been updated. (L 367)
- 385 – It should be “Conclusions”.
Response: Agreed, Text has been updated. (L 463)
- 393 – It should be initials instead of full names.
Response: Agreed, we have updated the text. (L 472-479)
References – Big mess! References must be in the MDPI style: abbreviations not full journal names, remove months, add DOI.
Response: Thank you for the suggestion, we have corrected all the references as suggested and as per the journal guidelines. Months have been removed, DOI has been added (L 500-650), but there were a few references that do not have a DOI identifier.
Dr. Bibhu Prasad Panda
Corresponding author
Reviewer 3 Report
Malik et al. have performed a study with Vitamin D-deficient Wistar rats which were administered a placebo preparation or a preparation containing Vitamin D from irradiated mushrooms or Vitamin D2 and D3 from commercial products. After 4 weeks the Vitamin D status, the bone health and also the expression of various genes were examined.
The whole manuscript is written in a fluid and excellent style. The study is planned in a very good way and the results are presented in detail.
HOWEVER, there are some issues which should be reconsidered by the authors:
Line 26: The abbreviation „VDR“ should be explained to the readership in the abstract. Please add the full term for „VDR“ (and „VDR“ in brackets). In return you can cancel the term „Vitamin D receptor“ in line 83 and only use „VDR“.
Lines 22, 78 and 80: please use either „25-OHD“, „25-OH-D“ or „25(OH) D“ or „25-hydroxyvitamin D“ throughout the whole text!
Line 100: please explain „IO farming“, especially the abbreviation „IO“!
Lines 100-104: This section on the mushrooms and Vitamin D supplements is far too short and thererfore incomplete. The previous publication by Malik et al., 2022, is not freely accessible in the internet. Therefore a more detailed description of the irradiation process (kind of UV-B source, duration, distance of the light source, preparation of the mushrooms etc.) is urgently be needed.
Please indicate whether the identity of the mushrooms were checked by an expert in this field or by using literature etc.
Furthermore the two supplements Lalmin and Calcirol should be further characterised (e.g. regarding the IU per tablet or capsule, batch number).
Line 106: The information whether male or female Wistar rats (or a mixutre of both sexes) were used for the experiments is urgently needed, as e.g. female hormones (see estrogens) might have as well an influence on the bone structures.
Lines 200 and 388/389: From the experiments performed it cannot be concluded in line 200 that „there was no general toxicity“, as no thorough toxicological examination of all relevant organs was performed. The only conclusion can be that there were no mortalities within the study period. In addition it is not possible to conclude that the intake of the irradiated mushrooms is safe (line 388). Furthermore the conclusion that the mushrooms in the rat models were „without any indication of toxicity“ (line 389) is not covered by a thorough toxiciological examinations of the relevant organs. Therefore these statements urgently have to be reconsidered and revised by the authors.
Line 204 / Figure 1: Please add that GP-1 did not receive vitamin D, but a placebo preparation
In addition the discussion of the results should also include following aspects:
1. All three irradiated mushrooms led to an increase of 25-hydroxyvitamin D serum levels, but there are differences in the Groups 2,3 and 4 regarding calcium, phophorous and ALP levels as well as differences concerning the expression of CYP isoenzymes and VDR in the liver and the kidney. Please add a discussion on these findings!
2. The mushrooms are irradiated resulting in a conversion of ergosterol to vitamin D2. However, some other structures/ingredients within the mushrooms might also change due to irradiation which might have an influence on the results of your study and also on the safety- Please discuss this aspect!
3. You tested three mushrooms. Please discuss, which of the mushrooms is most promising considering the price, the conversion rate of ergosterol to vitamin D2 (yield) and the pharmacological results obtained?
To sum up, this is a most interesting study with scientific significance. However, the issues mentioned should be addressed by the authors.
Author Response
Response to Reviewer 3
Comments and Suggestions for Authors
Malik et al. have performed a study with Vitamin D-deficient Wistar rats which were administered a placebo preparation or a preparation containing Vitamin D from irradiated mushrooms or Vitamin D2 and D3 from commercial products. After 4 weeks the Vitamin D status, the bone health and also the expression of various genes were examined.
The whole manuscript is written in a fluid and excellent style. The study is planned in a very good way and the results are presented in detail.
Response: We are thankful to the reviewer for recognising our hard work for this manuscript.
HOWEVER, there are some issues which should be reconsidered by the authors:
Line 26: The abbreviation „VDR“ should be explained to the readership in the abstract. Please add the full term for „VDR“ (and „VDR“ in brackets). In return you can cancel the term „Vitamin D receptor“ in line 83 and only use „VDR“.
Response: Thank you for the important suggestion. We have incorporated the suggestions in the revised manuscript (L 30)
Lines 22, 78 and 80: please use either „25-OHD“, „25-OH-D“ or „25(OH) D“ or „25-hydroxyvitamin D“ throughout the whole text!
Response: We have tried to maintain the unification of nomenclature as 25-OHD in the revised manuscript (25, 40, 91, 178, 256, 340, 342, 393, 398, 401).
Line 100: please explain „IO farming“, especially the abbreviation „IO“!
Response: IO is short for “Indian Organic” farming. We have updated full form in revised manuscript. (L 113)
Lines 100-104: This section on the mushrooms and Vitamin D supplements is far too short and thererfore incomplete. The previous publication by Malik et al., 2022, is not freely accessible in the internet. Therefore, a more detailed description of the irradiation process (kind of UV-B source, duration, distance of the light source, preparation of the mushrooms etc.) is urgently be needed.
Response: We appreciate the very important suggestion by the reviewer. We have described the irradiation process in the revised manuscript (L 119-126)
Please indicate whether the identity of the mushrooms were checked by an expert in this field or by using literature etc.
Response: The three mushrooms used in the study are most popular and largely cultivated varieties, thus are easy to identify. In addition, their identity was also confirmed from literature and by me (Dr B.P. Panda). Further the all the mushrooms sample were preserved in our laboratory.
Furthermore, the two supplements Lalmin and Calcirol should be further characterised (e.g. regarding the IU per tablet or capsule, batch number).
Response: We have added specifications as suggested. Both lalmin and calcirol were in powder form. The strength of Lalmin was 8k IU/g and Calcirol was 60k IU/g (L 115-117). For the study we calculated and weighed an amount that supplied 30IUs per dose (L 167-170).
Line 106: The information whether male or female Wistar rats (or a mixutre of both sexes) were used for the experiments is urgently needed, as e.g. female hormones (see estrogens) might have as well an influence on the bone structures.
Response: We used both male and female Wistar rats for our study, we have incorporated the information in the revised manuscript (L 146)
Lines 200 and 388/389: From the experiments performed it cannot be concluded in line 200 that „there was no general toxicity “, as no thorough toxicological examination of all relevant organs was performed. The only conclusion can be that there were no mortalities within the study period. In addition, it is not possible to conclude that the intake of the irradiated mushrooms is safe (line 388). Furthermore, the conclusion that the mushrooms in the rat models were „without any indication of toxicity “(line 389) is not covered by a thorough toxiciological examinations of the relevant organs. Therefore, these statements urgently have to be reconsidered and revised by the authors.
Response: As correctly mentioned, no thorough toxicological studies were carried out for the present study. This is a very important suggestion. This is surely a subject of future research which we stated in the conclusion remarks. We have revised the text and incorporated the changes as suggested (244-245, 466, 467, 470-471)
Line 204 / Figure 1: Please add that GP-1 did not receive vitamin D, but a placebo preparation
Response: We have already mentioned in the first place of methodology that Group 1 (GP-1) was maintained on a vitamin D deficient diet throughout the feeding period of 4-weeks and served as the control group of the experiment (L 167-168). As per authors opinion adding would be a repetition of same thing.
In addition, the discussion of the results should also include following aspects:
- All three irradiated mushrooms led to an increase of 25-hydroxyvitamin D serum levels, but there are differences in the Groups 2, 3 and 4 regarding calcium, phophorous and ALP levels as well as differences concerning the expression of CYP isoenzymes and VDR in the liver and the kidney. Please add a discussion on these findings!
Response: It is correct observation that all the three mushrooms led to an increase in 25-OHD levels, however there is certain difference in calcium, phosphorus and ALP levels as well as in the expressions of CYP isoenzymes and VDR in the liver and the kidney. The actual reason for this difference is not known, however, the possible variation may arise due to specie difference. In addition, the levels of PTH, calcium phosphorous and ALP are linked to the active metabolite of vitamin D i.e., 1,25-dihydroxyvitamin D, however 1,25-dihydroxyvitamin D was not analysed in this study and can be considered as a limitation. We have updated this in the discussion section (L 436-443)
The mushrooms are irradiated resulting in a conversion of ergosterol to vitamin D2. However, some other structures/ingredients within the mushrooms might also change due to irradiation which might have an influence on the results of your study and also on the safety- Please discuss this aspect!
Response: This is an important point to note, however an earlier study by Simon et al. (2013) have carried out an extensive study to investigate the suitability of UV light for vitamin D enrichment in mushrooms. They have concluded that compositional effects of UVB light are only limited to vitamin D and no such detrimental changes on the nutritional composition were identified. Further UVB exposure process for the production and sale of button mushrooms has been approved in European Union under novel food regulation (EC No. 258/97). In addition, few other studies have reported that UV exposure have negative effect on colour profile of edible mushrooms, however in our recent study (Malik et al., 2022), we observed that there was no such significant effect on the colour profile of edible mushrooms exposed to UVB for 120 mins (L 444-454).
You tested three mushrooms. Please discuss, which of the mushrooms is most promising considering the price, the conversion rate of ergosterol to vitamin D2 (yield) and the pharmacological results obtained?
Response: The finding of this study revealed that button mushroom is the most promising in terms of vitamin D yield (32.4 µg/g) as well as the pharmacological results. In addition, the price (per kg) of button mushroom is also on a lower side (200 INR) in comparison to oyster (350 INR) and shiitake (2200 INR). We have included the text in the discussion (L 455-462)
To sum up, this is a most interesting study with scientific significance. However, the issues mentioned should be addressed by the authors.
Response: We are thankful to the reviewer for inspiring us. We have thoroughly revised the manuscript, hope now it meets the journal standards.
Dr. Bibhu Prasad Panda
Corresponding author

Reviewer 4 Report
The topic of the manuscript is interesting because there is a global need of vitamin D fortified foods.
Moreover, fungi represent interesting sources of vitamin D, more sustainable than animal sources. In general, information on the bioavailability of vitamin D2 as compared to vitamin D3 is scarce, especially when vitamin D2 is produced after irradiation. Therefore, the manuscript provides new insights on the topic.
Nevertheless, the link between the current study and the possibility to use mushrooms as vitamin D sources needs to be explained in better details. One point to notice is that vitamin D2 is stable during storage in irradiated mushrooms upon drying and hence, mushrooms can represent a functional ingredient to produce vitamin D-enriched foods (see: Pedrali, D., Gallotti, F., Proserpio, C., Pagliarini, E., Lavelli, V. Kinetic study of vitamin D2 degradation in mushroom powder to improve its applications in fortified foods. LWT - Food Sci. Technol., 2020, 125, 109248).
As a second point, it should be noticed that formulation of mushroom to produce vitamin D enriched foods, opens up new challenges since the food matrix need to be optimized to maximize vitamin D bioavailability (see: Lavelli, V.; D’Incecco, P.; Pellegrino, L. Vitamin D incorporation in foods: Formulation strategies, stability, and bioaccessibility as affected by the food matrix. Foods 2021, 10, 1989).
In conclusion, I suggest to improve the discussion in order to emphasize the relevance of the study in the context of the worldwide deficiency of vitamin D.
Author Response
Response to Reviewer 4
The topic of the manuscript is interesting because there is a global need of vitamin D fortified foods.
Moreover, fungi represent interesting sources of vitamin D, more sustainable than animal sources. In general, information on the bioavailability of vitamin D2 as compared to vitamin D3 is scarce, especially when vitamin D2 is produced after irradiation. Therefore, the manuscript provides new insights on the topic.
Response: We are thankful to the reviewer for recognising our hard work for this manuscript.
Nevertheless, the link between the current study and the possibility to use mushrooms as vitamin D sources needs to be explained in better details. One point to notice is that vitamin D2 is stable during storage in irradiated mushrooms upon drying and hence, mushrooms can represent a functional ingredient to produce vitamin D-enriched foods (see: Pedrali, D., Gallotti, F., Proserpio, C., Pagliarini, E., Lavelli, V. Kinetic study of vitamin D2 degradation in mushroom powder to improve its applications in fortified foods. LWT - Food Sci. Technol., 2020, 125, 109248). As a second point, it should be noticed that formulation of mushroom to produce vitamin D enriched foods, opens up new challenges since the food matrix need to be optimized to maximize vitamin D bioavailability (see: Lavelli, V.; D’Incecco, P.; Pellegrino, L. Vitamin D incorporation in foods: Formulation strategies, stability, and bioaccessibility as affected by the food matrix. Foods 2021, 10, 1989).
Response: As correctly mentioned, vitamin D2 in irradiated mushrooms is stable during storage. Also, it is true that vitamin D (fat soluble vitamin) incorporation in foods is a challenging task. Several strategies have been employed in recent years (like nanoemulsion technology) to increase stability and bioavailability in food matrix. However, the authors would like share with the reviewer that in the present study we evaluated bioavailability of vitamin D from UV irradiated mushrooms and did not incorporate mushroom powder as a functional ingredient to produce any vitamin D enriched food product. However, this in an interesting point to note, where vitamin D enriched mushroom could be used as an active ingredient to fortify various food and this is for sure a subject of future research. Further we want to state that we have incorporated the statement regarding stability (L 455-456).
In conclusion, I suggest improving the discussion in order to emphasize the relevance of the study in the context of the worldwide deficiency of vitamin D.
Response: As suggested we have improved the discussion section (L 444-462)
We wish to thank all the reviewers for their time & efforts to provide useful suggestions that helped us to improve our manuscript.
Dr. Bibhu Prasad Panda
Corresponding author
Round 2
Reviewer 1 Report
Effect of Vitamin D Enriched Edible Mushrooms on Vitamin D Status, Bone Health and Expression of CYP2R1, CYP27B1 and VDR Gene in Wistar Rats
The Authors have made good progress on addressing my major concerns in the methods, results, and discussion. Some sentences still need some grammar improvement, but this can be corrected at a later stage. The following are all specific minor comments that can improve the manuscript.
Minor changes:
Consistency in spacing of the units throughout the manuscript. Note that lines 3,4, 129, 354 have different spacing. Please adjust the spacing for consistency throughout the manuscript.
Figures 4 and 5: Please add something to define the bar and error bars as done for Figure 2 (Add this information: each xxx represents the mean plus/minus xxx for n=x).
Line 47: Please remove the word “the”: should be… also report high levels of vitamin D…
Lines 89: Please consider removing “ have shown good bioavailability” or rewrite the sentence since it is awkward as written.
Line 89-91: Consider rewriting the sentence to something like: However, little is known about additional health-related benefits and the efficacy of UV-irradiated mushrooms to increase serum 25-OHD levels.
Line 93: Please change to: … this form of vitamin D is the major circulating….
Line 97: Please change to: The VDR protein binds to the active form….
Line 107: Please consider replacing observed to evaluated
Line 179, 181: For consistency, please consider 0- week. Meaning that the authors use 4-weeks so its best to be consist in how this is stated throughout the manuscript.
Author Response
Ans to Reviewer 1 Question
The Authors have made good progress on addressing my major concerns in the methods, results, and discussion. Some sentences still need some grammar improvement, but this can be corrected at a later stage. The following are all specific minor comments that can improve the manuscript.
Minor changes:
Consistency in spacing of the units throughout the manuscript. Note that lines 3,4, 129, 354 have different spacing. Please adjust the spacing for consistency throughout the manuscript.
Response: We want to thank reviewer for providing an opportunity to improve the manuscript. We have made the changes as suggested at (L 125, 128, 129, 355) and elsewhere in the text.
Figures 4 and 5: Please add something to define the bar and error bars as done for Figure 2 (Add this information: each xxx represents the mean plus/minus xxx for n=x).
Response: Suggestion has been added in the footnote of Figure 4 and 5 (L311 and 332)
Line 47: Please remove the word “the”: should be… also report high levels of vitamin D…
Response: Agree, Text has been updated (L47).
Lines 89: Please consider removing “have shown good bioavailability” or rewrite the sentence since it is awkward as written.
Response: Sentence has been rewritten (L87-89).
Line 89-91: Consider rewriting the sentence to something like: However, little is known about additional health-related benefits and the efficacy of UV-irradiated mushrooms to increase serum 25-OHD levels.
Response: Sentence has been rewritten (L89-91)
Line 93: Please change to: … this form of vitamin D is the major circulating….
Response: We have updated the text (L93)
Line 97: Please change to: The VDR protein binds to the active form….
Response: We have updated the text (L 97-98)
Line 107: Please consider replacing observed to evaluated
Response: Agree, we have updated the text (L 107)
Line 179, 181: For consistency, please consider 0- week. Meaning that the authors use 4-weeks so its best to be consist in how this is stated throughout the manuscript.
Response: As suggested, changes have been made (L 179-181).
We wish to thank the reviewer for his time & efforts to provide useful suggestions that helped us to improve our manuscript a lot.
Reviewer 2 Report
The authors corrected this paper properly taken under considerations all my comments. Therefore, I can accept it now.
Author Response
Answer to Reviewer 2 Question
The authors corrected this paper properly taken under considerations all my comments. Therefore, I can accept it now.
We wish to thank the reviewer for his time & efforts to provide useful suggestions that helped us to improve our manuscript.
Reviewer 3 Report
Malik et al. have submitted a revised manuscript based on the reviewers oppinions. In my point of view all suggestions and changes were performed in a very good way including the discussion part.
I have only two further (minor) points which need to be reconsidered by the authors:
1. Lines 115-117: The authors added the dosage of vitamin D2 and D3 and presented the IU values per gram in accordance to the declarations of these products which is correct. I propose also to add the values expressed in µg/g, i.e. 200 µg/g and 1500 µg/g, as the readership might not be so familiar with the conversion of IU of vitamin D to µg/g.
2. Lines 449-451: the Regulation EC No. 258/97 of the European Union is no longer valid and was superseded by the Regulation (EU) 2015/2283. In addition the reference needs to be changed as there is no information in the reference 58 concerning irradiated fungi. Thus the sentence and also the reference 58 urgently need to be changed:
In lines 449-551 it should be: „Further, the UVB exposure process for the production of button mushrooms has been approved in the European Union under the novel food regulation, i.e. Regulation (EU) 2015/2283 [58].“
In lines 646-648, reference 58 should be changed to:
„58. Commission implementing Regulation (EU) 2017/2470 of 20 December 2017 establishing the Union list of novel foods in accordance with Regulation (EU) 2015/2283 of the European Parliament and the Council on novel foods. Official Journal of the European Union No. L351, page 72, 30.12.2017. Link: https://eur-lex.europa.eu/legal-content/EN/TXT/PDF/?uri=CELEX:32017R2470&from=EN“
(remarks: the journal „Official Journal of the European Union“ can be abbreviated by „OJEU“. Please consider whether the link to the EU document should be added. In my point of view this might be a good idea as the document has no DOI and therefore the reader can easily look up this document)
Again, the study and the manuscript are well prepared. The results are scientific significance and of great interest to the readership of the journal.
Author Response
Answer to Reviewer 3 Question
Malik et al. have submitted a revised manuscript based on the reviewers opinions. In my point of view all suggestions and changes were performed in a very good way including the discussion part.
I have only two further (minor) points which need to be reconsidered by the authors:
- Lines 115-117: The authors added the dosage of vitamin D2 and D3 and presented the IU values per gram in accordance to the declarations of these products which is correct. I propose also to add the values expressed in µg/g, i.e. 200 µg/g and 1500 µg/g, as the readership might not be so familiar with the conversion of IU of vitamin D to µg/g.
Response: In the revised manuscript, we have expressed the values in IU/g as well as in µg/g as suggested (L115-117).
- Lines 449-451: the Regulation EC No. 258/97 of the European Union is no longer valid and was superseded by the Regulation (EU) 2015/2283. In addition the reference needs to be changed as there is no information in the reference 58 concerning irradiated fungi. Thus the sentence and also the reference 58 urgently need to be changed:
In lines 449-551 it should be: „Further, the UVB exposure process for the production of button mushrooms has been approved in the European Union under the novel food regulation, i.e. Regulation (EU) 2015/2283 [58].“
In lines 646-648, reference 58 should be changed to:
„58. Commission implementing Regulation (EU) 2017/2470 of 20 December 2017 establishing the Union list of novel foods in accordance with Regulation (EU) 2015/2283 of the European Parliament and the Council on novel foods. Official Journal of the European Union No. L351, page 72, 30.12.2017. Link: https://eur-lex.europa.eu/legal-content/EN/TXT/PDF/?uri=CELEX:32017R2470&from=EN“
(remarks: the journal „Official Journal of the European Union“ can be abbreviated by „OJEU“. Please consider whether the link to the EU document should be added. In my point of view this might be a good idea as the document has no DOI and therefore the reader can easily look up this document)
Response: As correctly suggested, we have updated the text and the reference in the revised manuscript (L 450-452). We agree that including a link will be helpful to easily find the document, so we have added the link. Official Journal of the European Union has been abbreviated as Off. J. Eur. Union in another mdpi journal article, (https://www.mdpi.com/2304-8158/9/5/665), So we have updated the reference accordingly (L647-650).
Again, the study and the manuscript are well prepared. The results are scientific significance and of great interest to the readership of the journal.
We wish to thank the reviewer for his time & efforts to provide useful suggestions that helped us to improve our manuscript a lot.
Dr Bibhu Prasad Panda
Corresponding Author
Reviewer 4 Report
The points raised by this reviewer have been addressed
Author Response
Answer to Reviewer 4 comments
The points raised by this reviewer have been addressed
We wish to thank the reviewer for his time & efforts to provide useful suggestions that helped us to improve our manuscript.
Dr Bibhu Prasad Panda
Corresponding author